# Mechanical Behavior of Different Restorative Materials and Onlay Preparation Designs in Endodontically Treated Molars

**DOI:** 10.3390/ma14081923

**Published:** 2021-04-12

**Authors:** Ana Beatriz Gomes de Carvalho, Guilherme Schmitt de Andrade, João Paulo Mendes Tribst, Elisa Donária Aboucauch Grassi, Pietro Ausiello, Guilherme de Siqueira Ferreira Anzaloni Saavedra, Adriano Bressane, Renata Marques de Melo, Alexandre Luiz Souto Borges

**Affiliations:** 1Department of Dental Materials and Prosthodontics, Institute of Science and Technology, São Paulo State University (Unesp), São José dos Campos 12245-000, Brazil; anab.gomes@hotmail.com (A.B.G.d.C.); guisdandrade@hotmail.com (G.S.d.A.); elisa.aboucauch@unesp.br (E.D.A.G.); guilhermesaavedra@icloud.com (G.d.S.F.A.S.); renata.marinho@unesp.br (R.M.d.M.); alexandre.borges@unesp.br (A.L.S.B.); 2Graduate Program in Dentistry, Department Dentistry, University of Taubate (UNITAU), Taubate 12020-270, Brazil; joao.tribst@gmail.com; 3Department of Neurosciences, Reproductive and Odontostomatological Sciences, School of Dentistry, University of Naples Federico II, 80138 Naples, Italy; 4Graduate Oral Health Applied Science, Institute of Science and Technology, São Paulo State University (Unesp), São José dos Campos 12245-000, Brazil; adriano.bressane@unesp.br

**Keywords:** mechanical stress, finite element analysis, dental onlays, dental prosthesis, dental materials

## Abstract

This study evaluated the effect of the combination of three different onlay preparation designs and two restorative materials on the stress distribution, using 3D-finite element analysis. Six models of first lower molars were created according to three preparation designs: non-retentive (nRET), traditional with occlusal isthmus reduction (IST), and traditional without occlusal isthmus reduction (wIST); and according to two restorative materials: lithium-disilicate (LD) and nanoceramic resin (NR). A 600 N axial load was applied at the central fossa. All solids were considered isotropic, homogeneous, and linearly elastic. A static linear analysis was performed, and the Maximum Principal Stress (MPS) criteria were used to evaluate the results and compare the stress in MPa on the restoration, cement layer, and tooth structure (enamel and dentin). A novel statistical approach was used for quantitative analysis of the finite element analysis results. On restoration and cement layer, nRET showed a more homogeneous stress distribution, while the highest stress peaks were calculated for LD onlays (restoration: 69–110; cement layer: 10.2–13.3). On the tooth structure, the material had more influence, with better results for LD (27–38). It can be concluded that nRET design showed the best mechanical behavior compared to IST and wIST, with LD being more advantageous for tooth structure and NR for the restoration and cement layer.

## 1. Introduction

Onlay is defined as “partial-coverage restoration that restores one or more cusps and adjoining occlusal surfaces or the entire occlusal surface and is retained by mechanical or adhesive means” [1]. Despite the excellent performance of onlays and metal crowns [2,3], the demand for materials that increasingly meets the aesthetic demand presses for the development of new materials. The advent of enamel etching in the mid-twentieth century [4], dentin adhesion in the 1970s [5], and the development of hydrofluoric acid etching of dental porcelains [6] made it possible to perform indirect adhesive partial restorations, such as laminate veneers, inlays, and onlays.

Partial-coverage adhesive restorations are a good alternative to full crowns on the treatment posterior teeth with great loss of dental structure [7,8,9]. Since onlay preparation limits are cavity-dependent [10], the preparation is less aggressive, removing less sound dental structure than a full crown [10,11,12]. The onlays are indicated when the tooth presents a great loss of coronary structure with less than 1.5 mm of thickness in the cusps; it is recommended to re-perform its covering, which can be total or partial, according to the coronary involvement [13,14]. Unlike onlays, restorations known as “tabletop” are indicated when there is structural loss due to erosion or wear in the occlusion, with indication for minimal or no preparation [15].

Dental materials technology evolved since the advent of partial adhesive restorations with the development of high-performance dental ceramics, indirect composite resin [16,17], CAD-CAM (Computer Aided Design-Computer Aided Manufacturing) systems [18], and reliable adhesive systems [19,20]. Thus, dental ceramics have become very popular; in addition, the evolution of these materials allows for the optimization of optical and mechanical properties of restorations, popularizing their use among clinicians and researchers [21,22,23]. Regarding restorative materials, ceramics and composites are the most used today for onlay manufacturing, as they present relatively compatible survival rates [24,25]. Composite resin seems to be indicated especially in patients with high masticatory forces and suspected parafunctional habits such as bruxism [26]. On the other hand, because they are more rigid, ceramic onlays have the ability to mimic tooth enamel, creating a restoration with mechanical behavior similar to the intact teeth [27].

In contrast, the concept of dental preparation has barely evolved [28,29]. The major differences between traditional ceramic and alloy preparations are that: the walls of the dental preparation must be more expulsive; box shapes may be present; grooves and sharp angles must be avoided; and bevels are not indicated [30]. However, with new materials and with the reliability of adhesive procedures, retention and resistance forms of cavity preparations can be questioned [27,28,29,31], since these forms are acquired at the expense of wear of sound dental structure.

The adhesive cementation process makes the restoration, adhesive layer, and tooth become a strong biomechanical unit, improving stress distribution [29]. Thus, some authors recently proposed new preparation designs for partial-coverage restorations, associating minimally invasive dentistry with a better force distribution in the remaining tooth structure [28,29,31,32,33]. Basically, for onlay restorations, it is possible to perform three different types of preparation: (1) Traditional onlay preparation, which is characterized by 6–10 degrees axial walls, rounded internal angles, at least 2 mm occlusal isthmus, proximal boxes, and chamfer cusp coverage [34]; (2) The Morphology Drive Preparation Technique [28], which is a preparation technique in which the occlusal reduction is guided by tooth morphology without isthmus preparation, and the preparation also contain butt-joint margins in the proximal box; (3) and Simplified Non-retentive Tooth Preparation, which consists of 2 mm occlusal reduction, a U-shaped proximal box with smooth transition, and an oblique bevel in the cavosurface angles [29].

For studies like this, Finite element analysis (FEA) recently demonstrated great relevance in dentistry, since it makes it possible to predict the mechanical and structural behavior of materials through a non-destructive and mathematical approach [26,35,36,37].

To date, few studies have evaluated the effect of the design of different full-coverage onlay preparations. Regarding the polymerization shrinkage stress, a study compared three types of preparations and concluded that less retentive preparations tend to reduce the polymerization shrinkage stress [38]. This same result was obtained in relation to fracture resistance. When comparing different preparation geometries, the teeth that presented both the least invasive and retentive preparations proved to be more resistant to fracture [39]; in addition, both lithium-disilicate and leucite-reinforced glass ceramic restorations appear to be promising for clinical application in preparations without retention forms [40]. For marginal adaptation, teeth with the most complex preparation design showed inferior marginal adaptation when compared to those with more simplified and less retentive geometries [41]. Clinically, the studies that evaluated the execution of restorative treatments with preparations without emphasis on the retention method were promising, recommending more and more the importance of the conservation of the remaining healthy dental structure [28], both for resin and ceramic restorations [10,11]. The guidelines for these preparations are introduced in critical literature reviews [29,31,34]; even so, it is advisable to carry out a long-term control of these restorations.

Based on this, the aim of this study was to evaluate the effect of preparation design and restorative material combination on stress distribution at the restoration, tooth structure, and adhesive interface, using the finite element analysis (FEA). The first null hypothesis is that the preparation design will not affect the stress distribution on restoration, cement layer, and tooth structure; the second null hypothesis is that the restorative material will not affect the analyzed structures.

## 2. Materials and Methods

The study followed a 3 × 2 factorial design, considering the factors’ preparation design: non-retentive adhesive preparation (nRET) [29], traditional all-ceramic onlay preparation with occlusal isthmus reduction (IST) [42], and traditional all-ceramic onlay preparation without occlusal isthmus reduction (wIST) [28,42]; and the onlay restorative material: lithium-disilicate (LD) and CAD-CAM nanoceramic resin (NR) (Figure 1).

To obtain the models for finite element analysis (FEA), all preparations were executed on a lower right first molar typodont (MOM, Marília, São Paulo, Brazil), according to the following recommendations: simplified non-retentive preparation (nRET)—occlusal reduction following the natural tooth morphology (2 mm on functional cups, 1.5 mm on non-functional cusp), no isthmus preparation, all angles and walls smoothed and rounded, U-shaped proximal box with smooth transition, and oblique bevel in the cavosurface angles [29,38]; traditional overlay with isthmus preparation (IST)—occlusal reduction following the natural tooth morphology (2 mm on functional cups, 1.5 mm on non-functional cusp), isthmus preparation of 2 × 2 mm, proximal box thickness of 1 mm, chamfer of 1 mm in the axial walls, and an overall preparation angle of 6–10° toward the occlusal aspect [38,43]; and traditional onlay without isthmus preparation (wIST)—butt-joint preparation in the proximal box with 1 mm of thickness, interior walls diverging 6–10°, occlusal anatomy reduction following fissure directions and the resulting proportion of the cusps (2 mm on functional cups, 1.5 mm on non-functional cusp), and chamfer preparation of 1 mm on the axial walls [28,38,42]. For the standardization of the preparation’s extension, a silicone matrix was made and used to mark the limits of extracoronary preparations with a brush pen.

An unprepared typodont and the prepared ones were digitally impressed with an intraoral scanner (CS 3600, Carestream, Nova York, NY, USA). The 3D “.stl” (file extension) mesh was exported to Non-Uniform Rational Basis Spline (NURBS) modeling software (Rhinoceros 6.0SR8, McNell North America, Seattle, WA, USA). Then, following the BioCAD technique [44], an intact tooth was designed. The root, pulp, and dentin morphology were estimated based on common tooth anatomy [45,46]. An endodontic treatment (crown-down technique with 4% conicity, 25% tapering), a large class II mesio-occluso-distal) (MOD) cavity with 2 ± 0.5 mm thickness of remaining wall, and a composite resin build-up were generated in order to simulate a clinical condition of tooth structure loss.

The 3D “.stl” mesh obtained from the digital impression of the preparations made in the typodont was used for modelling the preparation designs (Figure 2). Then, a Boolean subtraction was performed on the MOD restored tooth obtained by the BioCAD technique (Figure 3). The external layer of the onlay preparation was duplicated and used as a base for modelling a 100-micron cementing layer thickness. This step was repeated for each preparation design.

The geometries were imported into a Computer Aided Engineering (CAE) software (ANSYS 19.2, ANSYS Inc., Houston, TX, USA) in “.step” format, and tetrahedral elements were used to generate the mesh. The number of elements and nodes are described in Table 1, and they were defined after the mesh convergence test with 10% of relevance. The cervical region of the root (2 mm below the cement-enamel junction) was selected for system fixation condition, ensuring that the only movement constraint was on the *Z*-axis. The interfaces were considered perfectly bonded, and the geometries were considered isotropic, homogeneous, and linearly elastic. A vertical occlusal load of 600 N [47] was applied at the central fossa region, in the internal surface of the mesio and disto-lingual cusps, and in the internal surface of the buccal median cusp (tripod contact) (Figure 4) [48,49,50]. The solid volumes described in the final models are displayed in Table 1. The mechanical properties, such as elastic modulus (E) in GPa and Poisson’s ratio (V), were achieved through the literature or manufacturer data. The solid structures present in the final models consisted of: onlay made of lithium-disilicate (E = 95; V = 0.3) [51] or nanoceramic resin (E = 12.8; V = 0.3) [52], enamel (E = 84; V = 0.3) [53], dentin (E = 18; V = 0.23) [54,55], cement layer (E = 7; V = 0.3) [56], and bulkfill composite resin build-up (E = 8; V = 0.25) [57]. The results in the restoration, cement layer, and tooth structure were obtained using Maximum Principal Stress (MPS), which indicates the tensile stress results in MPa.

A limitation of the FE model may be the load application, since this analysis sim-ulates a static and not a dynamic load. In this study, only the application of axial load was considered to control the included variables in the present study. By applying loads on cusp inclination, the angle of the inclination could influence the results [54]. Another limitation is the fact that Residual shrinkage stress was not simulated, since it can influence the biomechanical behavior of the restoration and interfaces [58]. In addition, as it is an in silico analysis, it does not match all real clinical conditions, which is also a limitation of the technique.

A novel statistical approach was used for quantitative analysis of the finite element analysis results. For this, after the finite element analysis was performed, the tensile stress peaks on restoration; cementing layer and tooth structure were exported from the CAE software (ANSYS 19.2, ANSYS Inc., Houston, TX, USA), according to the element number corresponding to the numerical calculation. A correlation was made between real and theoretical probabilities, in order to define the distribution curve that best fits the data. The stress distribution was recorded as colorimetric maps (MPa) with adjustable color scale corresponding to the stress magnitude comparison between the preparation designs for each analyzed structure.

## 3. Results

The FEA results are represented in colorimetric graphs in Figure 5, and the values of the Maximum Principal Stress (MPa) in the form of distribution graphs are plotted in Figure 6. The shape parameters of the distribution graphs are summarized in Table 2. To obtain the stress distribution, the automatic labelling maximum value in the CAE software was used to detect the region of higher stress magnitude; in sequence, the stress data were exported in “.txt” file instead of colorimetric maps. The stress data were organized according to their distribution and shape. The stress on the tooth structure (enamel, dentine, and build-up) was measured using the Maximum probe detected by the Mechanical APDL (ANSYS 19.2, ANSYS Inc., Houston, TX, USA). After that, the peaks were plotted in bar graphs (Figure 6).

Observing the stress distribution on the restoration’s intaglio surface, LD onlays (69–110 MPa) showed higher tensile stress concentration than NR (10–24 MPa). For NR models, the wIST preparation stress peaks were twice that of nRET (12 MPa), with IST being slightly higher (17 MPa) than nRET. It can be seen in the FEA distribution graph (Figure 6) that stress peak values below 10 MPa were more frequent in the nRET LD and wIST LD groups. Stress values between 30 and 50 MPa were more frequent in the IST LD group, with a similar distribution for the nRET LD and wIST LD groups. For groups restored with NR, higher stress peaks were more frequent in the wIST group, followed by the IST and nRET groups. For LD onlays, the stress distribution on the restoration’s intaglio showed that the preparation design influenced the stress distribution; nRET had a homogeneous distribution and lower peaks (68 MPa), followed by IST (77 MPa), and higher for wIST (110 MPa) (Figure 5 and Figure 6).

## 4. Discussion

Results of this study indicated that the first null hypothesis was rejected, because the preparation design affected the stress concentration in the restoration, cement layer, and tooth structure.

The basic form of dental preparations did not significantly change over the years, even with the advent of new restorative materials [59]. However the mechanical behavior of restoration, cement layer surface, and tooth structure were affected according to the preparation design. The results of the present study confirm that non-retentive preparations have mechanical advantages in all analyzed structures.

The IST preparation was firstly designed on the non-adhesive restorations, presenting the concepts of mechanical retention and the material’s resistance [34]. However, with the development of the adhesive dentistry, these dental shapes are not required anymore. Moreover, the presence of shoulders and isthmus preparation provided a complex geometrical shape to the preparation, which promoted high stress concentration on all the simulated clinical situations observed in this study. In fact, retentive preparations with complex geometry have more internal angles, and these geometric changes result in greater stress in these areas, and these regions are potentially considered as breaking points for the restorations [21,40].

The lithium-disilicate wIST and IST groups concentrated more tensile stress on the restoration intaglio surface than nRET. Non-retentive preparations with simplified geometry can transform the negative tensile stresses for ceramic restorations into non-damaging compression stresses [27,29]. This aspect was also detected by Falahchai et al. [39] who found that simplified designs without retention forms reduced the incidence of restoration fracture.

Since the most frequent clinical failure pattern in teeth restored with onlay is fracture of the restoration [25,60,61], non-retentive preparation could increase the longevity of this type of restoration. Besides, the presence of pronounced shoulders was proven to require an extensive removal of tooth structure [62].

Another common clinical failure mode in onlays restorations is debonding [25]. Although it can be considered that more retentive preparations would have less risk of detachment or debonding, our study indicated that the nRET model preparation concentrated less tensile stress in the cement layer compared to the retentive one. In addition, the area with more stress concentration was located on enamel, a more reliable adhesive substrate [63]. In this sense, the occurrences of reported clinical failures could be associated with operative errors with the adhesive technique.

During the preparation design, IST preparation required more tooth reduction on tooth structure, resulting in loss of structural tissue, while wIST could reduce the loss of dental tissues [28]. Thus, the execution of isthmus preparation could weaken the dental structure [64], especially because the intracoronal extension can create a wedge effect [39]. In this sense, the nRET and wIST preparations were more advantageous for dental structure integrity.

The nRET preparation does not require resistance and retention forms; the geometry follows a smooth and fluid curve with open angles. These characteristics render this operative technique easier to perform, and it also provides a minimally invasive intervention once there is no need to remove sound tooth structure to achieve the ideal geometric forms. It was specified that dental procedures, especially inlays and onlays, performed by unexperienced professionals tend to present higher failure rates when comparing to experienced dentists [65].

On the other hand, the retentive features of IST and wIST will provide a defined path of insertion of the onlay, which will facilitate seating during cementation and reduce the exposure of the cement at the margin [25]. Besides that, the longevity of the restorative treatment in non-retentive preparation relies on the adhesion to dental tissues and restorative material. Thus, techniques such as immediate dentin sealing [66], air abrasion [67], oblique cut of the enamel [68], and the use of reliable adhesive materials supported by the literature are indicated [69].

The second null hypothesis of the study, that the restorative material would have no effect on mechanical behavior, was also rejected. That is, the results showed that LD onlays concentrate more tensile stress on the restoration and in the cement layer, while NR onlays concentrate stress mainly in the tooth structure.

When a more elastic material was used, the design of the preparation did not significantly affect the mechanical behavior of the restoration. Composite resin onlays can be advantageous, especially in patients with high masticatory forces and suspected parafunctional habits such as bruxism, since there is a lower risk of the restoration fracture [26,49,50]. Additionally, when comparing the mechanical behavior of resin-based materials, especially for the nRET preparation, this material presented a mechanical behavior that resembles a natural tooth, as analyzed by Costa et al., 2017 [70], which evaluated the influence of different occlusal contacts and used premolars as models.

Resin-based materials (NR) homogeneously distributed stress n almost all of the cement surface, with a higher frequency of lower values, while lithium-disilicate (LD) concentrated higher peaks in more localized points (at the preparation margins and at the axial-occlusal edge) (Figure 5). Since the bond strength of lithium-disilicate is higher than that of nanoceramic resin, the highest stress values for LD are not critical for the occurrence of debonding. The same premise is valid for NR. Since the peaks in FEA were between 8 and 9.6 MPa, they do not reach 50% of the microtensile bond strength value of the composite resin [71]. Given this assumption, composite resin-based onlays could be cemented over non-retentive preparations [49,50].

On the other hand, NR onlays promoted a higher stress concentration and higher peaks in the dental structure. Therefore, in situations with extremely fragile teeth, with thin remaining walls or with the presence of cracks, the restorative material with the greatest biomechanical advantage seems to be dental ceramics [27,72]. Based on what was exposed, it is assumed that despite the better mechanical performance of NR, which acts as a stress absorber due to the different Young’s modulus than LD (NR: E = 12.8; LD: E = 95), both materials have their own clinical indication, and clinical success can be achieved using both materials, depending on the correct indication.

A possible limitation of the study is that the simulated mesio-occluso-distal cavity was designed arbitrarily with the CAD software, and not made from a real clinical condition such as those simulated in patient-specific FEA studies. Probably, a patient-specific assessment could bring new information about the effect of the preparation design [26,73]. In addition, the simulated occlusal contact generates loads in the axial direction [74]; horizontal loads could simulate more critical effects on tooth, cement layer, and restoration. Further studies should evaluate the clinical variables, such as the effect of fatigue behavior, accuracy, and precision of digital and traditional impression, and clinical trials should be encouraged to better understand the effect of the preparation design.

## 5. Conclusions

Within the limitations of this study, it is possible to appreciate that:The finite element analysis carried out on non-retentive onlay dental preparation showed the best mechanical behavior compared to other preparation designs;The finite element analysis also showed that resin-based materials presented a better mechanical behavior than lithium-disilicate ceramic;Lithium-disilicate ceramic materials could represent an interesting alternative of restorative material in specific clinical situations, such as extremely fragile teeth or in the presence of cracks.

## Figures and Tables

**Figure 1 materials-14-01923-f001:**
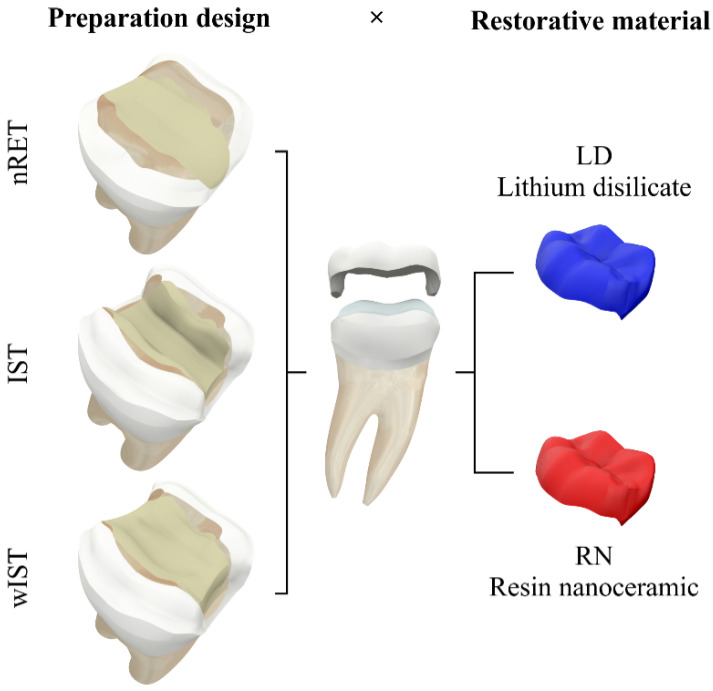
Study groups. Factor preparation design: nRET—Non retentive adhesive preparation; IST—traditional all-ceramic onlay preparation design with occlusal isthmus reduction; wIST—traditional all-ceramic onlay preparation without occlusal isthmus reduction. Factor onlay restorative material: lithium-disilicate (LD); nanoceramic resin (NR).

**Figure 2 materials-14-01923-f002:**
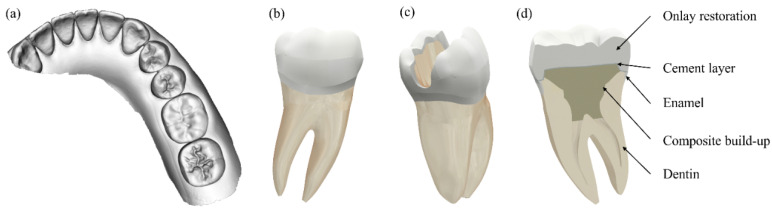
(**a**) Typodont digital impression of the intact tooth; (**b**) digital modelling in NURBS of the intact tooth with anatomic dental structures; (**c**) MOD cavity simulation; (**d**) Structures of the final model.

**Figure 3 materials-14-01923-f003:**
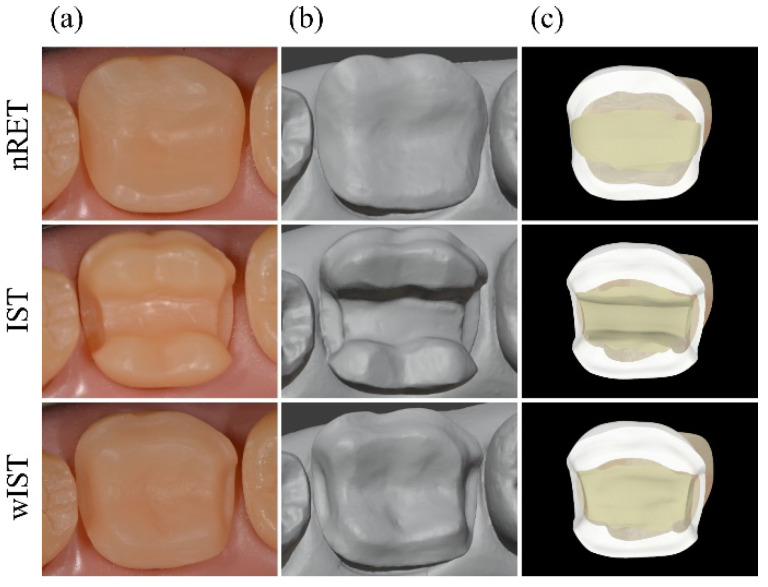
Steps for obtaining FEA preparations models: (**a**) onlay dental preparation on typodont; (**b**) virtual model obtained by digital impression; (**c**) designing of preparations using BioCAD technique.

**Figure 4 materials-14-01923-f004:**
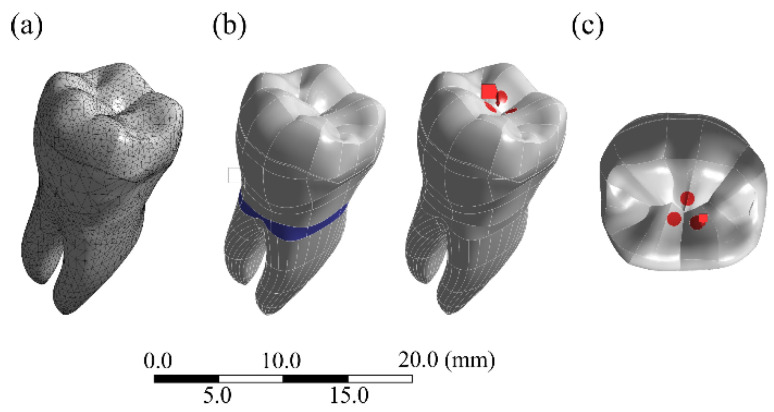
FEA processing steps: (**a**) mesh generation; (**b**) fixation of the system; and (**c**) axial load application (600 N).

**Figure 5 materials-14-01923-f005:**
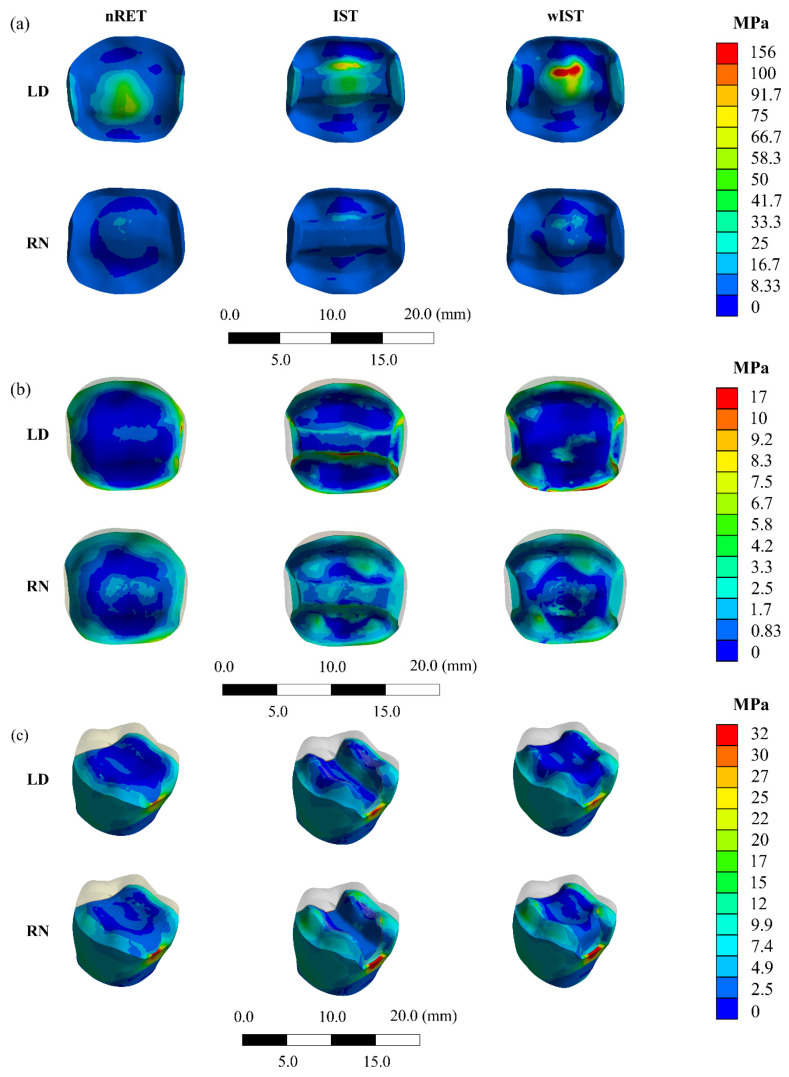
FEA Maximum Principal Stress results: (**a**) restoration’s intaglio surface; (**b**) cement layer; (**c**) tooth structure.

**Figure 6 materials-14-01923-f006:**
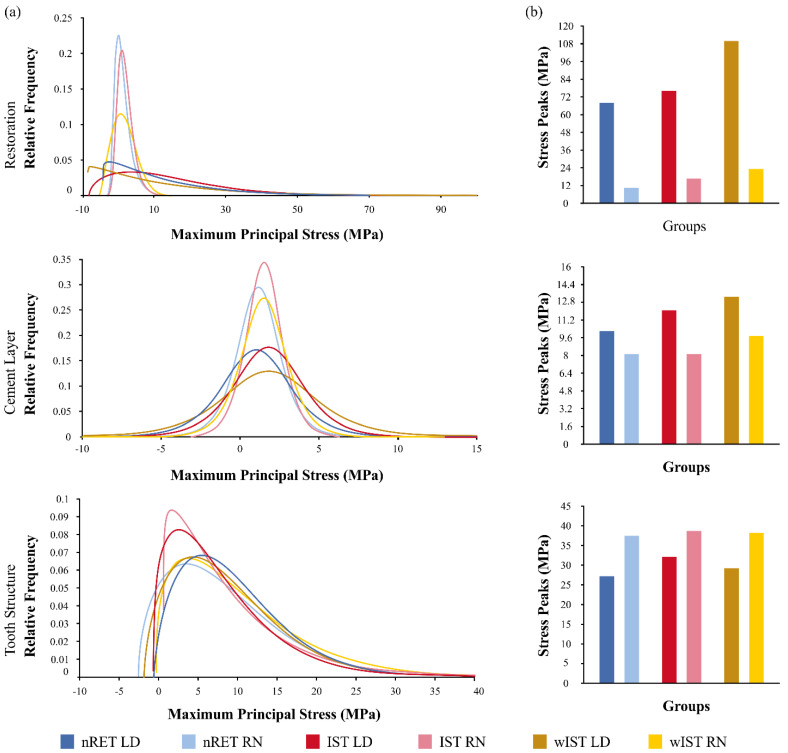
Quantitative FEA analysis: (**a**) distribution graph of the stress data on the restoration exported from the analysis software; (**b**) stress peaks on each group.

**Table 1 materials-14-01923-t001:** Number of nodes, elements, and volume of tooth structure reduction (sum of the restoration and cement layer volume), according to the model of each preparation design.

Preparation Design	Nodes	Elements	Volume (mm^3^) *
nRET	196,316	160,426	159
IST	199,025	160,944	179
wIST	200,391	161,748	159

* Sum of the dentin volume and enamel of each preparation design.

**Table 2 materials-14-01923-t002:** Distribution and shape parameter of each of the stress peak data of each group.

Structure	Group	Distribution	Correlation Coefficient	Shape/* Local	Scale
Restoration	nRET LD	Weibull	0.917	1.51	16.13
	nRET NR	GEV	0.841	−2.46 *	1.63
	IST LD	Weibull	0.948	1.89	20.64
	IST NR	GEV	0.848	2.96 *	1.49
	wIST LD	Weibull	0.918	1.48	19.75
	wIST NR	Weibull	0.962	2.80	7.40
Cement Layer	nRET LD	Weibull	0.968	2.01	11.06
	nRET NR	Weibull	0.913	2.16	11.05
	IST LD	Weibull	0.976	1.71	9.14
	IST NR	Weibull	0.973	1.49	8.36
	wIST LD	Weibull	0.949	2.26	11.45
	wIST NR	Weibull	0.967	1.54	10.79
Tooth Structure	nRET LD	Logistic	0.973	1.03 *	1.36
	nRET NR	Logistic	0.996	1.09 *	0.85
	IST LD	Logistic	0.969	1.79 *	1.36
	IST NR	Logistic	0.992	1.53 *	0.72
	wIST LD	Logistic	0.966	1.76 *	1.84
	wIST NR	Logistic	0.994	1.51 *	0.91

* GEV: generalized extreme value.

## Data Availability

Data sharing is not applicable for this article.

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
