# Peer review of "Mechanical Behavior of Different Restorative Materials and Onlay Preparation Designs in Endodontically Treated Molars"

_materials, 2021, doi:10.3390/ma14081923_

Round 1

Reviewer 1 Report

General comments

This laboratory study aimed to investigate the effect of 3 onlay preparation designs and 2 restorative materials on the stress distribution under simulated loading using finite element analysis.

Title: consider to specify the study was on “root-treated molar”

Introduction:

  1. Please rearrange the introduction to improve the flow, after the definition of the onlay, please add 1-2 lines on the problems of traditional onlays (Ln 67-69). Then talk about the advent of adhesive technology (Ln50-52) followed by the advantages of adhesive onlays over traditional onlays (Ln54-57).
  2. 56: “wearing” is not an appropriate word
  3. Summarize the existing evidence of similar studies, highlight the deficiencies and justify the need of performing this study.

Materials and methods:

  1. Please justify the location of loading point. Apart from the central fossa, buccal cusp of the mandibular molar is also considered to be the primary loading point during function.
  2. Table 1: as mentioned by the authors in the introduction, the amount of tooth structure removal for different designs are different, why were the volumes of tooth structure similar among the 3 designs?
  3. Please specify how/where the stress on the tooth structure (enamel/dentine) was measured. Why was the composite-build up which was a big bulk supporting the onlay not measured?

Results:

  1. Pleases specify “tooth structure”.

Discussion:

  1. Ln210: If the results showed the preparation design affected the stress distribution, the null hypothesis should be “rejected” not “accepted”.
  2. Ln247: ‘wear” is not an appropriate term. Consider to change it to “required more tooth reduction”
  3. Minimal invasive onlay is getting popular for restoring vital teeth with occlusal tooth wear or cracked tooth instead of root-treated teeth. Therefore, the FE model of using composite base is restricted to root-treated teeth only not applicable to other clinical situations.

Author Response

Please, Sir, see the attachment

Reviewer 2 Report

the current study investigates the effect of combining three onlay preparation designs and two restorative materials on the loading response and distribution using finite element modelling. The authors develop several models according to tradition techniques and theories to simulate the clinical condition. The authors obtained the max stress for evaluating the analysed loading applied.

The title is too long and does not reflet what has been done in the study, make the title simple and clear.

The abstract is poorly written, it was difficult to understand what exactly was done in this study until reaching line 34.

Remove lines 30-34 from the abstract this is an unnecessary detailed information which can be added in the materials and method section

I find adding 9 authors in a 13 pages manuscript brings a questions of who did what exactly in this study!

Line 36 “the results (MPa)” what does the authors means by this it seems this sentence was misplaced here

I think in simple words the authors did an FE study to study the stressed applied on the restoration, cement layer and tooth structure. Please try to workout this for the title and also make the abstract more clear it is really confusing to read.

Line 41 “promising mechanical behavior,” is not a good conclusion to mention, it is very generic and does not give any indication about the study findings.

Please consider reviewing the abstract and highlight the novelty, major findings and conclusions.

Introduction is very short and does not provided detailed literature analysis of past studies and what they did and how does the current study brings new knowledge to field.

In the introduction answer the following questions, what is the research gap did you find from the previous researchers in your field? Mention it properly. It will improve the strength of the article.

Change section 3 name to results and discussion and remove section 4 (suggestion)

Combine smaller paragraphs into larger ones.

Line 216-217 like what? Expand on this generic conclusion you need to give more details

Line 222-226 this is generic comments what is the benefit of mentioning this here.

What are the limitation in your FE models? Please mention them clearly in your materials and method section

The results are merely described and is limited to comparing the experimental observation. The authors are encouraged to include a more detailed discussion section and critically discuss the observations from this investigation with existing literature.

Line 303-308 should be moved to the introduction it doesn’t belong here at the end

Conclusion should be expanded upon and improved

Author Response

Please, Sir, see the attachment

Round 2

Reviewer 1 Report

The authors addressed most of my comments except Q.4.  I could neither find the changes specified (Ln174-180) nor any explanations why only the central fossa was used as the loading point. 

Author Response

Please, see the attached file. Thank you

Reviewer 2 Report

all questions answered 
